# Technological Properties of Acetylated Pigeon Pea Starch and Its Stabilized Set-Type Yoghurt

**DOI:** 10.3390/foods9070957

**Published:** 2020-07-18

**Authors:** Aderonke Olagunju, Olufunmilayo Omoba, Victor Enujiugha, Adeola Alashi, Rotimi Aluko

**Affiliations:** 1Department of Food and Human Nutritional Sciences, University of Manitoba, Winnipeg, MB R3T 2N2, Canada; monisola.alashi@umanitoba.ca (A.A.); rotimi.aluko@umanitoba.ca (R.A.); 2Department of Food Science and Technology, Federal University of Technology, Akure, Ondo State 340001, Nigeria; osomoba@futa.edu.ng (O.O.); vnenujiugha@futa.edu.ng (V.E.)

**Keywords:** pigeon pea starch, heating behaviour, yoghurt, cold storage, syneresis

## Abstract

The behaviour of graded acetylated pigeon pea starch during heat processing was evaluated in addition to the corresponding effect of their incorporation at 1.5% (*w*/*v*) as a stabilizer in set-type yoghurt. Acetylated starch possessed higher solubility and swelling power than native starch under the temperature regimes considered. Addition of acetylated pigeon pea starch as a stabilizer in yoghurt had positive influence on the water holding capacity (7.7% to 10.4% compared to 13.3% in yoghurt stabilized with native pigeon pea starch) and whey syneresis (approximately 15%, 12%, and 8% increase observed in yoghurt with acetylated pea starch compared to 47% in yoghurt with native pea starch stabilizer) at the end of a 28-day cold storage period. In addition, pea starch-stabilized yoghurt possessed an enhanced sensory attribute (firmness), and compared favourably with gelatin-stabilized yoghurt in terms of overall acceptability. Thus, acetylated pigeon pea starch exhibited improved physicochemical properties and showed usefulness as a stabilizer in yoghurt because it enhanced the physicochemical, storability, and sensorial quality, while improving the body and texture of the product.

## 1. Introduction

Yoghurt is a product of the lactic acid fermentation of milk and it is among the most common fermented dairy product widely consumed in many parts of the world. Syneresis (expulsion of whey) is a common technological defect in cultured semi-solid products, especially during refrigerated storage [1]. Control of syneresis has been facilitated by use of food additives to stabilize products as well as contribute to texture and viscosity [2]. Stabilizers are added to yoghurt to improve product stability and consumer acceptance; they are commonly used in fermented products to enhance texture and reduce wheying off. Reduced whey separation is achieved by binding water in order to reduce water flow in the yoghurt matrix; this imparts good syneresis resistance and results in a smooth mouth feel during consumption [3]. Stabilizers are applied to yoghurts to confer appropriate texture; vegetable gum, gelatin, carrageenan, sodium alginate, and starch are the commonly used stabilizers in yoghurt [4,5,6,7]. However, starch is the most commonly used stabilizer in food systems because it is cheaper, readily available, and confers technological properties.

Native starches have some limitations preventing their use for industrial applications; these limitations can be overcome by techniques developed to modify, and thus, improve the physicochemical properties of starch. Modified food starches, specifically from cassava and corn, have been used as stabilizers in food systems to confer specific functions, such as increased viscosity, improved mouthfeel and creaminess, and reduced syneresis of yoghurt [8,9]. Several researchers have reported the application of cereal or tuber starches as stabilizers in yoghurt manufacture [9,10,11,12,13]. Legume starches have superior functional properties in terms of solubility, swelling, and retrogradation compared to cereal or tuber starches [14]. However, there is limited information on the use of legume starches (native or modified) as yoghurt stabilizers.

Pigeon pea (*Cajanus cajan*) is a legume that forms a good portion of the human diet in many African, Asian, and South American countries [15]. The legume is often used as food and forage; the seed is rich in starch, protein, carbohydrate, and mineral elements [16]. In Nigeria, pigeon pea seeds used to be a delight and delicacy about two decades ago, but presently are an underutilized legume tending toward extinction, which makes it less popular than other food legumes like cowpea and soybeans [17]. The under-utilization and under-cultivation are attributable to scarce information regarding the nutraceutical and functional properties of the seed components (especially protein and starch fractions). Relevant and useful research output in the public domain may create awareness of the functional diversity of pigeon pea seeds, promote consumption and cultivation of the crop, while preventing overdependence on other food crops.

Our recent research has focused on evaluating the protein and starch components of pigeon pea seed. The antioxidant and antihypertensive properties of enzymatic hydrolysates and ultrafiltered peptides of pigeon pea have been reported [18,19]. Additionally, food grade starch was successfully extracted from pigeon pea seeds; the starch was subjected to different levels of acetylation to produce modified starch with potential application as an ingredient in food systems based on the evaluated physicochemical properties [20]. The current study, therefore, seeks to evaluate the functionality of acetylated pigeon pea starches when subjected to heat processing, while exploring the suitability of pea starch as a stabilizer in set-type yoghurt.

## 2. Materials and Methods

Native and acetylated pigeon pea starches obtained from a previous study [19] in our laboratory were used. PPNS (native pigeon pea starch) was earlier subjected to graded acetylation, producing starch with different degrees of substitution (DS) viz., 0.05, 0.09, and 0.14, thus, PPAS_1_ (acetylated pigeon pea starch DS 0.05), PPAS_2_ (acetylated pigeon pea starch DS 0.09), and PPAS_3_ (acetylated pigeon pea starch DS 0.14) were obtained and used in this study. Starter culture DANISCO Yo Mix 495 LYO (containing *Streptococcus thermophilus* and *Lactobacillus delbrueckii* subsp. *bulgaricus*) was obtained from the Dairy unit, University of Manitoba, Winnipeg, Manitoba, Canada. Commercial homogenized and pasteurized milk (3.5% fat, 3.3% protein, 13% total solids, pH 6.7) was purchased from Maizube farms, Minna, Niger State, Nigeria; skimmed milk powder (SMP) and sucrose were purchased from Ceci Supermarkets, Alagbaka, Akure, Ondo State, Nigeria. All reagents were of analytical grade and procured from Fisher Scientific (Oakville, ON, Canada) and Sigma Chemicals (St. Louis, MO, USA).

### 2.1. Determination of Functional Properties of Starch

#### 2.1.1. Determination of Solubility and Swelling Power

The solubility and swelling power of starch was determined, following the method of Gani et al. [21]. Suspensions (1%, *w/w*) of starch were prepared in a flask, heated to 50, 60, 70, 80, and 90 °C for 30 min, with shaking every 5 min, and left for cooling to room temperature, thereafter, being centrifuged for 15 min at 3000× *g*. The supernatant was decanted, and the residual volume was determined. The solid part was dried in an oven for 2 h at 130 °C.

#### 2.1.2. Determination of Paste Clarity

Paste clarity/light transmittance of starch paste was determined, as described by Craig et al. [22]. An aqueous suspension of starch (1% *w*/*v* db) was heated in a water bath at 90 °C for 30 min with constant agitation to ensure complete pasting. The starch paste was cooled to room temperature (around 25 °C) and the light transmittance of pastes was measured at 650 nm. The samples were stored for 5 days at 4 °C in a refrigerator and percentage transmittance was determined every 24 h using a UV/visible spectrophotometer (Ultrospec 4300 pro, Cambridge, England).

#### 2.1.3. Determination of Pasting Properties of Acetylated Starches

The pasting property was determined using a Rapid Visco Analyser (RVA) (Newport Scientific, Warriewood, Australia), as described in AACC 44-31A [23]. About 2.5 g of the sample was weighed into a dry empty canister and 25 mL of distilled water was dispensed into the container. The solution was thoroughly mixed, and the canister fitted into the RVA, according to the manufacturer’s instructions. The slurry was heated from 50 to 95 °C with a holding time of 2 min, and then, cooled to 50 °C with 2 min holding time. The rates of heating and cooling were maintained constant at 11.25 °C/min. Peak viscosity, trough, breakdown, final viscosity, set back, peak time, and pasting temperature were obtained from the pasting profile with the aid of Thermocline for Windows Software (TCW, v. 2.3, Warriewood, Australia).

### 2.2. Preparation of Set-Type Yoghurt

The yoghurt was prepared in triplicate, as described by Cui et al. [8]. Two litres of commercial homogenized and pasteurized milk were fortified with 2% (*w*/*v*) skimmed milk powder (SMP); 2% (*w*/*v*) sucrose was added as sweetener, followed by addition of 1.5% (*w*/*v*) acetylated or native pigeon pea starch as the stabilizer, and then, mixed thoroughly. The mixture was pasteurized at 85 °C for 30 min, cooled to 40–43 °C, and starter culture (DANISCO Yo Mix; 50 mg/L of yoghurt mix, 20 Danisco culture units (DCU) per 100 L) was added, then, distributed into sterile 250 mL containers to allow easy collection of sample for analysis. Fermentation was carried out at 40 °C for 6 h to develop proper acidity. Fermentation was terminated and the yoghurt samples were stored at 4 °C for 24 h prior to chemical analysis, sensory evaluation, and storage stability test. The prepared yoghurt was stored in the refrigerator (4 °C) for 28 days and samples (three independent containers of 250 mL) were analysed for each of the parameters evaluated at 7-day intervals throughout the refrigerated storage period. A commercial yoghurt sample was used as a control, while laboratory control was prepared with the above stated materials (similar to experimental sample) and 0.2% gelatin as the stabilizer.

### 2.3. Chemical Attributes of Yoghurt

#### 2.3.1. Acidity

pH of the yoghurt was measured at 23 °C using a pH meter (Accumet AB150, Thermo Fisher Scientific, Singapore) after calibrating with fresh pH 4.0, 7.0, and 10.0 standard buffers.

#### 2.3.2. Determination of Titratable Acidity (TA)

TA expressed as percent of lactic acid was determined using titrimetric method, as described by Schmidt et al. [10]. Yoghurt samples were warmed to 23 °C in a water bath and mixed well with a stirring rod. Then, about 10 g sample were transferred into a 50 mL beaker, 20 mL distilled water was added, and they were gently mixed together. The mixture was titrated against 0.1 M NaOH solution using phenolphthalein as the indicator to an end-point of faint pink colour. The titratable acidity was calculated using the equation below:$$\mathrm{Acidity}\;(\%)=\frac{\mathrm{Titre}\;\mathrm{value}\;\left(\mathrm{mL}\right)\times M\times 90\times 100}{\mathrm{Sample}\;\mathrm{wgt}\;\left(g\right)\times 1000}$$
where M is molarity of NaOH and 90 is equivalent molecular weight of lactic acid.

#### 2.3.3. Determination of Total Solids (TS)

Yoghurt samples were mixed thoroughly using a glass rod and about 1 g of sample was oven-dried for 2 h; the percentage Total Solids (% TS) was calculated using the equation below:$$\%\;\mathrm{TS}=\frac{\mathrm{wgt}\;\mathrm{of}\;\mathrm{sample}\&\mathrm{petridish}\;\mathrm{after}\;\mathrm{heating}\;\left(g\right)-\mathrm{wgt}\;\mathrm{of}\;\mathrm{empty}\;\mathrm{petridish}\;\left(g\right)\times 100}{\mathrm{Sample}\;\mathrm{wgt}\;\left(g\right)}$$

#### 2.3.4. Determination of Syneresis

Syneresis of yoghurt samples was measured according to the modified method of Guzman-Gonzalez et al. [24]. Approximately 5.0 g stored yoghurt sample (28 days at 4 °C, samples taken every 7 days) was placed in a pre-weighed tube and centrifuged for 10 min at 3000× *g* and 4 °C. The amount of supernatant whey was recorded. The percentage syneresis was calculated using the equation below:$$\mathrm{Syneresis}\;(\%)=\frac{\mathrm{Total}\;\mathrm{weight}\;\mathrm{of}\;\mathrm{drained}\;\mathrm{whey}\;\left(g\right)}{\mathrm{Total}\;\mathrm{weight}\;\mathrm{of}\;\mathrm{yoghurt}\;\left(g\right)}\text{×}100$$

#### 2.3.5. Determination of Water Holding Capacity

The water holding capacity (WHC) of yoghurt is defined as its ability to hold all or part of its own water. WHC of the samples was determined using method reported by the authors in reference [25]. Yoghurt sample (20 g) was centrifuged at 1250× *g* at 4 °C for 10 min. The supernatant was collected and weighed, and WHC was calculated according to the following equation:$$\mathrm{WHC}\;(\%)=\frac{\mathrm{weight}\;\mathrm{of}\;\mathrm{yoghurt}-\mathrm{weight}\;\mathrm{of}\;\mathrm{expelled}\;\mathrm{whey}\;}{\mathrm{weight}\;\mathrm{of}\;\mathrm{yoghurt}}\text{×}100$$

### 2.4. Sensory Evaluation of Yoghurt

Assessment of the sensory attributes was carried out as described by Soukoulis et al. [26]. Thirty panelists experienced in the evaluation of fermented dairy products were chosen for the assessment of the sensory attributes of yoghurt samples. Flavour, texture, and appearance of the samples were evaluated. A predetermined list of seven (7) sensory attributes was used to describe the sensory characteristics of yoghurts. A 2 h training session was conducted to evaluate the use of the attributes by the panelists during sensory analysis. The sensory attributes allowed the differentiation of samples in terms of appearance (colour), texture (firmness, consistency, syneresis), flavour (flavour and aroma), taste (palatability), and overall acceptability. Samples were coded using a 3-digit random number and served to the panelists for independent evaluation; all sensory attributes assessed by the panelists were rated using a 9-point hedonic scale (1 = extremely dislike, 9 = extremely like). The descriptions for evaluation of sensory attributes of yoghurt adapted from Soukoulis et al. [26] are shown in Table 1. The sensory evaluation was carried out in accordance with guidelines for human studies approved by the Ethical Committee of School of Agriculture and Agricultural Technology, Federal University of Technology, Akure, Ondo State, Nigeria (approval number FUTA/SAAT/2017/013).

### 2.5. Statistical Analysis

Data were generated in triplicate and subjected to Analysis of Variance (ANOVA) using Statistical Package for Social Sciences (SPSS) V. 17.0 (IBM Inc., New York, NY, USA). The mean values were separated using Duncan’s Multiple Range Test (DMRT) at 95% confidence level.

## 3. Results

### 3.1. Effect of Acetylation on Solubility and Swelling Power of Pigeon Pea Starch

The effect of temperature on solubility of native and acetylated pea starches was progressive (Figure 1A). The percentage solubility of the acetylated starches increased with an increase in temperature higher than those observed in native starch (3.5% at 50 °C to 20.9% at 90 °C). Starch with the highest DS (PPAS_3_) also had the highest starch solubility (5.7% at 50 °C to 34.6% at 90 °C). Similarly, acetylation increased the swelling power, which is a measure of the hydration capacity of the starch granules. The swelling power showed direct proportionality to temperature and degree of acetylation.

### 3.2. Effect of Storage Period on Light Transmittance of Native and Acetylated Pigeon Pea Starches

The effect of acetylation and storage duration on paste clarity of pigeon pea starch is shown in Figure 1B. The light transmittance ranged from 28% to 72% at 0 h and 10% to 68% at 120 h; however, values for native and acetylated pigeon pea starch gels decreased after 120 h of refrigerated storage. The light transmittance of native (PPNS) starch gel decreased from 28% to 10% after 120 h of refrigerated storage, while those of acetylated starch gels decreased from 41%, 63%, and 72% to 28%, 59%, and 67% for PPAS_1_, PPAS_2_, and PPAS_3_, respectively.

### 3.3. Pasting Properties of Native and Acetylated Pigeon Pea Starch

Pasting properties of pigeon pea starch as influenced by acetylation are summarized in Table 2. All parameters evaluated showed significant changes; reduced pasting properties were observed in the acetylated starches, which may be due to the introduction of the bulky acetyl group that weakened starch–starch interactions. In this study, pasting temperature (PT) ranged from 86 to 90 °C. PPNS had the highest pasting temperature, however, acetylation significantly (*p* < 0.05) reduced PT (87.80 to 85.60 °C). Pasting time (Pt) ranged from 4.93 to 5.07 min. The acetylated starches required less time to form a paste when compared to the native starch. Peak viscosity (Pv) ranged from 314.70 to 588.70 RVU and values reduced with an increase in level of acetylation. Fv of starches followed similar trend as Pv and Tv with PPNS showing highest Fv and the value decreased with increase in DS of starch. Similar to other pasting parameters, native pea starch showed highest breakdown viscosity (BD) and setback (SB), which was significantly (*p* < 0.05) reduced by 24% to 49% and 38% to 44%, respectively, when the starch was acetylated.

### 3.4. Physicochemical Properties of Yoghurt Stabilized with Acetylated Pigeon Pea Starch

Acetylated pigeon pea starch was employed as the stabilizer in set-type yoghurt and the physicochemical properties presented in Table 3. pH of yoghurt samples ranged from 4.30 to 4.63, while acidity (% lactic acid) was 0.9 to 1.2. Solids content was higher in yoghurt with stabilizer (18.2% to 20.5%) than 14.9% in the yoghurt with no stabilizer (YoCN). Yoghurt samples showed significant differences (*p* < 0.05) in percentage syneresis, with yoghurt stabilized with native pea starch (YoPN) exuding the highest amount of liquid from the yoghurt matrix, thus, having the highest degree of syneresis (46%). The degree of substitution showed an inversely proportional relationship with degree of syneresis; the yoghurts containing acetylated pea starch had lower syneresis (18.6% to 22%) compared to YoCN and YoPN. However, the market yoghurt (CTR) had a far lower level of syneresis at 10.8%.

### 3.5. Effect of Acetylated Pigeon Pea Starch as Yoghurt Stabilizer during Refrigerated Storage

Water holding capacity (WHC) and whey syneresis of the yoghurt during the 28 days was evaluated at 7-day intervals throughout the refrigerated storage period (Figure 2A,B). Commercial yoghurt (CTR) possessed superior WHC, which was almost steady within the first 21 days of refrigeration storage (94–96%). Modified pigeon pea starch also exhibited the ability to promote WHC (76–85%), and invariably reduced whey syneresis (18–25%). Percentage syneresis in CTR, LCTR, YoPA1, YoPA2, and YoPA3 remained almost unchanged throughout the 28 days storage period, whereas YoCN and YoPN showed higher syneresis, which was initially steady in the first 14 days of refrigerated storage (46.5% to 50.6% and 36.8% to 38.5%, respectively), but increased afterwards to 74.5% and 54.2%, respectively, on the 28th day.

### 3.6. Sensory Attributes of Yoghurt Stabilized with Pigeon Pea Starch

The sensory attributes of the laboratory control (gelatin-stabilized yoghurt) compared favourably with the market control (commercial sample), except for aroma, flavour, and overall acceptability (Table 4). Generally, all the yoghurt samples (CTR, LCTR, YoCN, YoPN, YoPA1, YoPA2, and YoPA3) showed no significant difference (*p* > 0.05) for colour (7.1 to 7.6), whereas firmness of the yoghurt was significantly different (*p* < 0.05) from one another (5.8 to 7.2).

## 4. Discussion

Increased solubility in acetylated starches can be attributed to the increased hydrophilicity of the starch [27]. Solubility of starch is an indicator of the degree of starch granule dispersion after cooking [28]. Similar to our observation, Wani et al. [27] reported that acetylation of black gram starch resulted in increased starch solubility. The increased solubility could also imply increased amylose leaching from swollen starch granules [29].

The swelling behaviour of starch is dependent on the amylose content, as well as the presence of amylopectin and non-carbohydrate substances, especially lipids, which may inhibit swelling. Swelling commenced at 60 °C earlier than observed in native starch and the highest swelling capacity was observed at 90 °C for all starches (2.5 to 4.0 g/g), which indicates that the penetrating power of water into starch granules can be increased at high temperature. The increase in swelling power may be a result of starch gelatinization, which causes disruption of the granular membrane and weakening of the internal starch structure that favoured increased water absorption. Acetylation conferred a 1.5 to 2.1-fold increase in swelling of starch paste, an effect that progressed with temperature. Introduction of the bulky acetyl group promotes structural reorganization, resulting in repulsion between starch molecules, which facilitates increased water percolation within amorphous regions of starch granules with a consequent increase in swelling capacity [30].

The high solubility and swelling capacities of acetylated pigeon pea starch may also be due to the introduction of hydrophilic substituting groups that allows the retention of water molecules, owing to their ability to form hydrogen bonds. This ensures a high retention of the water that enters the granule, thereby increasing the swelling capacity. This is a useful property in the manufacture of some confectionery products. A similar trend showing increased swelling capacity has been reported for chickpea [31] and black gram [27] starches after acetylation.

PPAS_2_ and PPAS_3_ exhibited higher light transmittance than PPNS and PPAS_1_. More so, PPNS and PPAS_1_ displayed significant decreases in light transmittance up until the fifth day, unlike PPAS_2_ and PPAS_3_ that exhibited lower rates of decrease in light transmittance during the storage period. Low light transmittance observed in native pigeon pea starch may be a consequence of turbidity development during storage caused by granule swelling and intermolecular bonding that enhanced retrogradation of amylose chains [32]. In addition, the significant decrease in light transmittance of native starch may be associated with the reassociation of gelatinized starch molecules (amylose, amylopectin) into an ordered crystalline structure [33]. Higher light transmittance after acetylation may be attributed to the introduction of the acetyl group to replace the hydroxyl group in the starch molecule [27]. Substitution of hydroxyl groups with acetyl groups may cause repulsion between adjacent starch molecules, hence, reducing inter-chain association, which enhanced light transmittance. Our reports corroborate the findings of Wani et al. [27], where acetylated black gram starches had higher light transmittance than native starch. Ayucitra [30] also reported increased starch paste clarity after acetylation of corn starch.

Pasting temperature (PT) is an important indicator of the minimum temperature required to achieve cooking in the starch sample and depicts the temperature at which viscosity begins to increase during the heating process. In this study, PT ranged from 86 to 90 °C. PPNS had the highest pasting temperature, however, acetylation significantly (*p* < 0.05) reduced PT (87.80 to 85.60 °C). The high pasting temperature observed in PPNS indicate native pigeon pea starch granules exhibited higher resistance to swelling and membrane rupture, while lower pasting temperature in acetylated starches may be as a result of the introduction of the acetyl group to the amorphous region, which made swelling of the acetylated starch easier than in the native starch. The decreased pasting temperature may be a result of weakening of starch structure due to interrupted hydrogen bonds upon introduction of the acetyl group. This property is important in systems where thickening of gels is required at low temperature in order to reduce energy cost during production [34].

Native pigeon pea starch required more time to form a paste, whereas acetylated starches required less. Pasting time (Pt) is a measure of the cooking time [35] and this indicates the duration taken for the starch to reach highest viscosity.

Peak viscosity is the maximum viscosity developed during or soon after the heating portion of the pasting period and it correlates to the final product quality. Generally, high viscosity of starch is desirable for industrial uses, for which a high thickening potential at high temperature is required and a high peak viscosity suggests more starch has been gelatinized during processing. The peak viscosity of native pigeon pea starch was higher than the modified counterpart, suggesting that the starch can be incorporated into food that requires high thickening such as custard, while lower viscosity of the acetylated starches could facilitate better flow of starch paste when introduced into food systems. Similar to the present finding, Wani et al. [27] reported a decrease in peak viscosity for acetylated black gram starches. Likewise, Wojeicchowski et al. [36] observed a Pv decrease in carioca bean starch after acetylation (DS 0.064). However, a contrary result of increased peak viscosity was reported by Wani et al. [37], after acetylation of Indian kidney bean. Trough viscosity (Tv) was lower in the acetylated starches than the native pea starch and the reduction progressed with an increase in DS of the acetylated starches.

Final viscosity (Fv) is a parameter used to indicate the ability of starch paste to form a viscous paste resultant from retrogradation of soluble amylose during cooling and gelling of amylose [38]. High Fv, as observed in native pea starch, suggests that the paste is highly resistant to mechanical shear, which may, in turn, produce a more rigid or firm gel, hence, a higher tendency to retrograde upon cooling due to recrystallization of leached amylose molecules. However, lower Fv of modified starches suggests they may produce a less rigid gel when compared to the native starch. Similar to our observation, Wani et al. [27] reported reduced Fv for Indian black gram starch after acetylation.

Breakdown (BD) measures the tendency of swollen starch granules to rupture when held at higher temperatures and subjected to continuous shearing; this is also indicative of the stability of the starch upon heating as well as a measure of the susceptibility of cooked paste to disintegration. Highest breakdown viscosity (BD) was observed in PPNS; acetylation, however, significantly (*p* < 0.05) reduced this property by 24% to 49%. High BD infers less stability of starch during heating. This property was lower in acetylated starches, which suggests that they may exhibit better starch granule swelling with slow tendency to lose viscosity and might be able to withstand more heating and shear stress [34]. Aryee et al. [39] reported that high paste stability is an indication of very weak cross-linking within the starch granules, which implies that such starch cannot be used for products where starch stability is required at very high temperature due to its tendency to breakdown. Low BD values, as observed for the acetylated starches, suggest stability of starches under hot conditions. The results from this study infers that acetylated starches, especially the PPAS_3_, will be more stable under high temperature processing or storage conditions than PPNS. High BD values reflect granular swelling that make starch granules more susceptible to shear [40], thus, low BD as observed in acetylated starches, infers they have better resistance to shear and hence, less susceptible to granular disintegration.

Set back (SB) of the starches was significantly (*p* < 0.05) lower in the acetylated starches (202.4 to 222.80 RVU) compared to the native pigeon pea starch counterpart. Ashogbon [41] reported a low SB (154.29 RVU) for bambara starch when compared to cereals. Similarly, low set back viscosity was reported for acetylated black gram starches in comparison to the native counterpart [27]. The results suggest that the acetylated starches may be less susceptible to retrogradation, which agrees with a previous report where acetylated pigeon pea starches exhibited lower retrogradation tendency than native pea starch [19]. Setback (SB) involves the rearrangement of starch molecules and shows the tendency of starch molecules to retrograde [42], thus, lower setback during the cooling of the paste indicates lower tendency for retrogradation [43].

Acidity of yoghurt is a result of fermentation of milk sugar (lactose) into lactic acid by the starter culture during incubation. The development of lactic acid in fermented dairy products is essential for the proper formation of a yoghurt gel network. Acidity is inversely related to pH value, as increased lactic acid production results in a corresponding pH drop. Nevertheless, pH is a valuable tool for measuring the extent of lactic acid fermentation, since acidity influences the perception of other attributes [44] and plays an important role in yoghurt flavour [45]. pH ranged from 4.30 to 4.63 and was inversely proportional to yoghurt acidity (0.90% to 1.21% lactic acid). The low pH is attributable to the fact that yoghurt is a fermented product. Control yoghurt (market sample) had the highest pH, followed closely by gelatin-stabilized yoghurt, while samples with pigeon pea starch had lower pH with no significant difference (*p* > 0.05) between them. This suggests that incorporation of pea starch had no significant effect on pH.

The acidity of commercial yoghurt (0.90%) was significantly different from those of developed samples (1.03% to 1.21%), and this may be due to differences in starter culture, milk type, fermentation time, and temperature. Acidity observed for LCTR was within the range (0.98% to 1.10%) in another report for gelatin-stabilized camel milk yoghurt [46], which may be due to the higher pH of the yoghurt, likewise, variation in the type of milk (influencing the lactose available for fermentation) used as well as variation in the starter culture employed in market sample and experimental yoghurts.

Solids content was higher in yoghurt with stabilizer (pigeon pea starch, gelatin or commercial stabilizer) than the YoCN, whereas graded acetylation of pigeon pea starch showed no effect on the solid content of yoghurt. The high total solids content supports the firm, custard-like body of set-type yoghurt; values are significantly higher than the 13% to 17% range earlier reported by Smit [47] for commercial yoghurts.

Syneresis ranged from 10.8% to 46.5%, and the market sample (CTR) exhibited the lowest value; this property significantly differed among the yoghurt samples. The loss of ability of the yoghurt gel to entrap the serum phase due to the weak gel network results in whey separation, which is also referred to as syneresis [48]. The percentage of syneresis of prepared yoghurts showed positive correlation with the earlier discussed physicochemical properties of PPAS_3_ in terms of retrogradation tendency, breakdown, and set back. The improved properties of PPAS_3_ may have contributed to the improved ability to bind the free water as well entrap the yoghurt gel during the coagulation process. Significant whey syneresis (46.5%) observed in YoCN may be attributed to the absence of a stabilizer and, hence, the expulsion of liquid could not be efficiently prevented. The yoghurt stabilized with acetylated starches had lower syneresis values (19–22%), although was significantly higher than the gelatin-stabilized yoghurt and market sample. However, acetylated starch with higher DS (PPAS_2_ and PPAS_3_) resulted in reduced whey expulsion when compared to 36.8% in the sample stabilized with native starch (YoPN). Overall, starch-stabilized yoghurts showed 53–60% reductions in whey syneresis when compared to control yoghurt without pea starch (YoCN). Syneresis is an important factor in set-type yoghurt, as it can lead to the accumulation of whey on the surface of the yoghurt, which can result in poor consumer acceptance [49]; the present result may have influenced the sensory results.

The DS of the pea starch showed no significant effect on the WHC of yoghurt; however, native pea starch exhibited lower WHC (66%), which gradually reduced over the 28-day cold storage period. The WHC (85%) of yoghurts with stabilizer (acetylated starches) was significantly (*p* < 0.05) higher than yoghurt without pigeon pea starch that had 57% WHC at day 0. More so, the WHC of YoPA_1_, YoPA_2_, and YoPA_3_ was not significantly different (*p* > 0.05) from each other and compared with no significance difference with LCTR. There were no significant changes in the WHC of each yoghurt sample during storage, except on the 28th day, where lower WHC for some of the samples was observed. The results suggest that introduction of acetylated pea starches as stabilizers was able to effectively bind water, thus, reducing water loss from the yoghurt gel matrix.

The stabilized yoghurts showed no significant susceptibility to syneresis over the refrigerated storage period when compared to yoghurts with no stabilizer (YoCN) or with native starch (YoPN), which was initially stable for 14 days, after which progressive syneresis was observed. Nonetheless, YoPN had a lower degree of serum separation than YoCN, suggesting that native starch exhibited better ability to retain water within the yoghurt matrix than samples without stabilizer. Degree of substitution had no significant effect on the amount of syneresis, as yoghurts with the acetylated starches (YoPA_1_, YoPA_2_, and YoPA_3_) showed no significant difference (*p* > 0.05) in syneresis percentage. Although the acetylated pigeon pea starches did not totally prevent serum separation during storage, they sufficiently reduced syneresis to levels that are similar to that of the commercial sample.

The sensory attributes of the laboratory control (gelatin-stabilized yoghurt) compared favourably with the market control (commercial sample) except for aroma, flavour, and overall acceptability (Table 3); this may be attributable to the fact that the stabilizers used in the market sample differ from those in experimental samples and were inclusive of fortifying agents (protein concentrate—as indicated on the label). LCTR compared closely with commercial yoghurt for most of the sensory attributes evaluated. Gelatin is an extensively employed stabilizer known to improve the textural and sensorial qualities of yoghurt, which evidently was close to qualities of the commercial yoghurt. However, pea starch showed the ability to perform in close capacity as gelatin; the use of different levels of acetylated starch as a stabilizer did not produce any significant difference (*p* > 0.05) in their sensory attributes, as shown by the similar values obtained for YoPA_1_, YoPA_2_, and YoPA_3_. Moreover, the colour of all yoghurt samples was not significantly different (*p* > 0.05) from the commercial sample, thus, addition of pigeon pea starch had no effect on the colour quality. The introduction of PPAS_3_ in yoghurt showed the best stabilizing effect by enhancing firmness of the sample (YoPA_3_); regardless, this sample did not significantly differ (*p* > 0.05) from the others, except YoCN, which may result from the absence of any form of stabilizing agent in the product.

The sensory results correlate to the WHC and syneresis, where YoPA_3_ also showed superior properties. The sensory ratings showed that the use of native starch was inadequate for yoghurt stabilization because YoPN exhibited poor firmness and high syneresis. YoPA_3_ had the highest overall acceptability among the experimental yoghurt samples, which might be attributed to the superior firmness. The lower level of syneresis may have contributed to the increased firmness of YoPA3, despite the lower viscosity of the modified starch ingredient. This is because texture (firmness, consistency, and syneresis) is one of the major determinants of yoghurt quality based on its influence on yoghurt appearance, mouthfeel, and overall acceptability [50]. Earlier, superior sensory scores were reported for yoghurt containing gelatin than those of carboxymethylcellulose and corn starch [51]. Similarly, Soomro et al. [52] reported higher sensory quality for milk yoghurt stabilized with gelatin than those stabilized with pectin.

The higher sensory scores for firmness, consistency, syneresis, and overall acceptability showed that the use of acetylated pigeon pea starch as a yoghurt structure stabilizer has greater potential than the native starch with respect to adoption by the industry as a food ingredient.

## 5. Conclusions

Acetylation of pigeon pea starch enhanced the physicochemical properties (increased solubility, swelling, and reduced pasting properties), which promoted its suitability as a functional ingredient in yoghurt formulation. The use of acetylated starch as a stabilizer was comparable with gelatin and suitable for significant reductions in serum separation, while maintaining firmness, consistency, flavour, and overall acceptability of yoghurt. PPAS_3_ with 0.14 DS produced the most acceptable sample and could be considered a potential ingredient for industrial production of set-type yoghurt.

## Figures and Tables

**Figure 1 foods-09-00957-f001:**
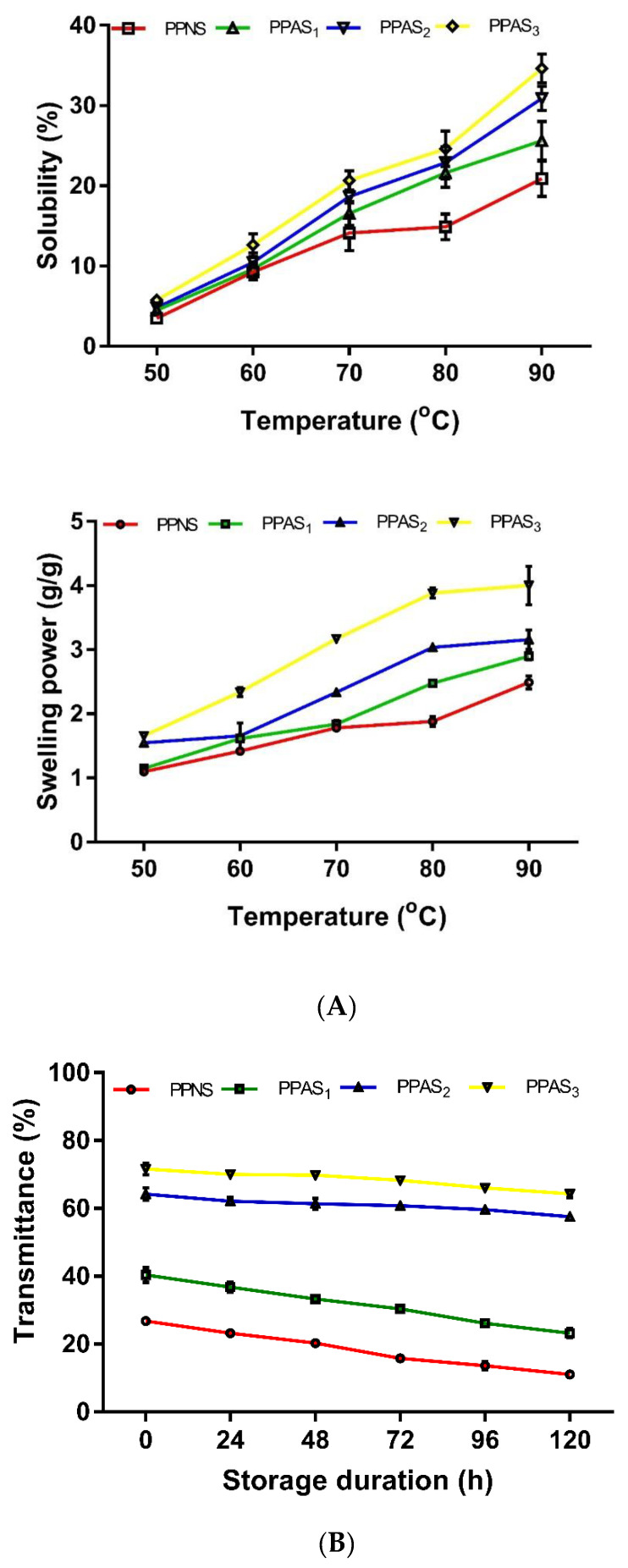
(**A**) Effect of temperature on solubility and swelling power of native and modified pigeon pea starches. PPNS—Pigeon pea native starch; PPAS_1_—Pigeon pea acetylated starch, DS 0.05; PPAS_2_—Pigeon pea acetylated starch, DS 0.09; PPAS_3_—Pigeon pea acetylated starch, DS 0.14. (**B**) Effect of storage duration on transmittance of native and modified pigeon pea starches. PPNS—Pigeon pea native starch; PPAS_1_—Pigeon pea acetylated starch, DS 0.05; PPAS_2_—Pigeon pea acetylated starch, DS 0.09; PPAS_3_—Pigeon pea acetylated starch, DS 0.14.

**Figure 2 foods-09-00957-f002:**
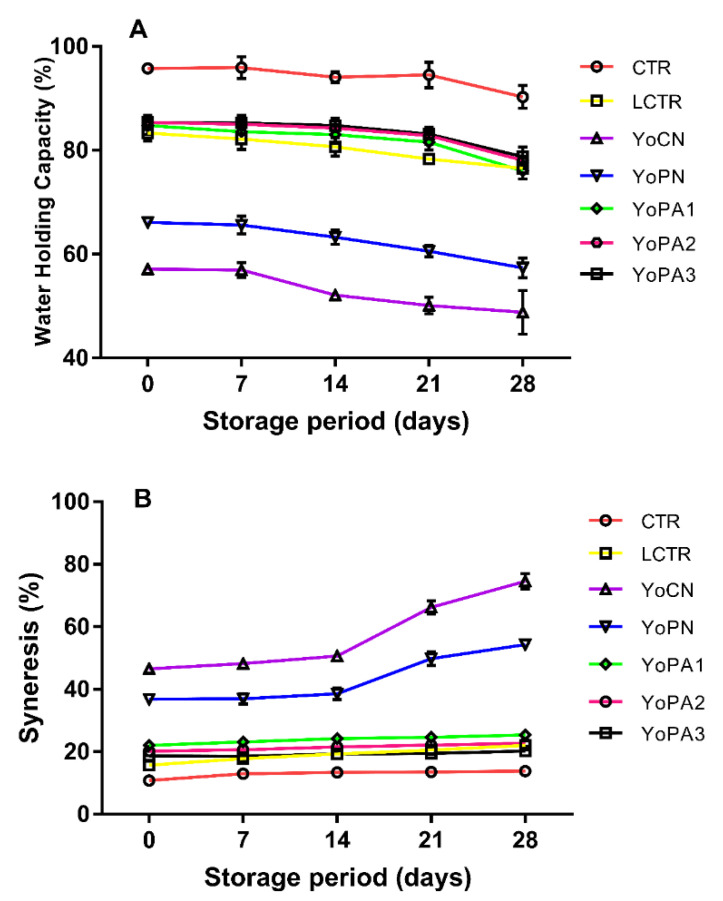
(**A**,**B**) Effect of refrigeration storage on water holding capacity and syneresis of yoghurt. CTR—Commercial yoghurt; LCTR—gelatin-stabilized yoghurt; YoCN—yoghurt without stabilizer; YoPN—yoghurt stabilized with native starch; YoPA1—yoghurt stabilized with DS 0.05 acetylated starch; YoPA2—yoghurt with stabilized with DS 0.09 acetylated starch; YoPA3—yoghurt stabilized with DS 0.14 acetylated starch.

**Table 1 foods-09-00957-t001:** Definition of the descriptors (sensory attributes) used for evaluation of the pigeon pea acetylated starch stabilized set-type yoghurts.

Attribute	Description
Aroma	Detect any aroma defects (e.g., unclean, masked, unnatural, cooked, lacks freshness) by smelling and oral perception of samples
Colour	Assess the colour (white, whitish, yellowish, yellow) of yoghurt
Palatability	Evaluate the taste of samples considering several attributes associated with taste (e.g., Unclean, unnatural, whey and refreshing perception) and aftertaste (e.g., sourness, astringency, sweetness, bitterness, and saltiness)
Firmness	Estimate the hardness, brittleness, gumminess, and gelatin-like texture of the coagulum
Consistency	Evaluate the consistency when stirring the product with the spoon, determine the rheological behaviour of yoghurt in the mouth
Flavour	Evaluate the flavour (pleasant, palatable or unpleasant)
Syneresis	Visual observation of the yoghurt surface; examine whey drainage after inserting the spoon into the curd
Overall acceptability	Rate the overall score of the sample considering the appearance, taste, and flavour profiles.

**Table 2 foods-09-00957-t002:** Pasting properties of native and acetylated pigeon pea starch.

Properties	PPNS	PPAS_1_	PPAS_2_	PPAS_3_
Pasting temp (°C)	89.95 ± 0.71 ^a^	87.81 ± 0.51 ^b^	87.40 ± 0.23 ^c^	85.60 ± 0.17 ^d^
Pasting time (min)	5.07 ± 0.11 ^a^	5.05 ± 0.10 ^a^	5.00 ± 0.10 ^a^	4.93 ± 0.20 ^a^
Peak Viscosity (RVU)	588.70 ± 0.83 ^a^	408.50 ± 0.51 ^b^	346.88 ± 0.99 ^c^	314.70 ± 1.27 ^d^
Trough Viscosity (RVU)	434.80 ± 1.12 ^a^	293.90 ± 0.83 ^b^	282.44 ± 1.07 ^c^	263.50 ± 0.92 ^d^
Final Viscosity (RVU)	794.40 ± 2.55 ^a^	576.70 ± 1.60 ^b^	503.10 ± 3.42 ^c^	465.90 ± 1.91 ^d^
Break Down (RVU)	153.90 ± 0.91 ^a^	116.72 ± 0.83 ^b^	114.80 ± 0.86 ^b^	78.20 ± 1.11 ^c^
Set Back (RVU)	359.60 ± 1.61 ^a^	222.80 ± 0.91 ^b^	223.10 ± 0.82 ^b^	202.40 ± 0.63 ^c^

Values are means ± standard deviation of three determinations. Values with different letters (superscript) on the same row are significantly different (*p* ≤ 0.05). PPNS—Pigeon pea native starch; PPAS_1_—Pigeon pea acetylated starch, DS 0.05; PPAS_2_—Pigeon pea acetylated starch, DS 0.09; PPAS_3_—Pigeon pea acetylated starch, DS 0.14.

**Table 3 foods-09-00957-t003:** Physicochemical properties of yoghurt stabilized with acetylated pigeon pea starch.

Sample	pH	Acidity (% Lactic Acid)	Total Solids (%)	Syneresis (%)
CTR	4.63 ± 0.11 ^a^	0.90 ± 0.01 ^c^	18.53 ± 0.34 ^b^	10.80 ± 1.26 ^g^
LCTR	4.41 ± 0.05 ^b^	1.03 ± 0.01 ^b^	18.20 ± 0.20 ^b^	15.69 ± 0.55 ^f^
YoCN	4.31 ± 0.00 ^c^	1.21 ± 0.01 ^a^	14.86 ± 1.21 ^c^	46.54 ± 0.23 ^a^
YoPN	4.30 ± 0.01 ^c^	1.17 ± 0.06 ^a^	20.20 ± 0.62 ^a^	36.80 ± 0.09 ^b^
YoPA1	4.32 ± 0.00 ^c^	1.10 ± 0.01 ^ab^	20.41 ± 0.50 ^a^	22.00 ± 0.11 ^c^
YoPA2	4.31 ± 0.00 ^c^	1.21 ± 0.08 ^a^	20.33 ± 0.76 ^a^	20.18 ± 0.20 ^d^
YoPA3	4.31 ± 0.00 ^c^	1.21 ± 0.01 ^a^	20.47 ± 0.23 ^a^	18.62 ± 1.63 ^e^

Values are mean ± standard deviation of three determinations. Values with different letters (superscript) on the same column are significantly different (*p* ≤ 0.05).

**Table 4 foods-09-00957-t004:** Sensory attributes of yoghurt stabilized with acetylated pigeon pea starch.

	Samples						
Attributes	CTR	LCTR	YoCN	YoPN	YoPA1	YoPA2	YoPA3
Aroma	7.80 ± 1.61 ^a^	6.65 ± 0.85 ^b^	5.90 ± 0.99 ^b^	6.20 ± 1.03 ^b^	6.20 ± 1.03 ^b^	6.70 ± 1.34 ^b^	6.10 ± 0.87 ^b^
Colour	7.50 ± 1.58 ^a^	7.50 ± 1.20 ^a^	7.10 ± 0.84 ^a^	7.40 ± 0.84 ^a^	7.50 ± 0.70 ^a^	7.60 ± 0.70 ^a^	7.50 ± 0.97 ^a^
Palatability	7.90 ± 1.28 ^a^	7.30 ± 0.90 ^a^	5.30 ± 1.03 ^b^	5.80 ± 0.94 ^b^	5.70 ± 1.63 ^b^	5.90 ± 2.02 ^b^	6.00 ± 1.25 ^b^
Firmness	6.90 ± 1.81 ^ab^	7.00 ± 1.50 ^ab^	5.80 ± 1.03 ^b^	6.10 ± 0.99 ^ab^	6.20 ± 1.47 ^ab^	6.20 ± 0.37 ^ab^	7.20 ± 1.13 ^a^
Consistency	8.00 ± 1.24 ^a^	7.30 ± 1.50 ^b^	5.80 ± 1.23 ^c^	6.00 ± 0.94 ^c^	6.50 ± 1.34 ^c^	6.30 ± 1.56 ^c^	6.30 ± 1.33 ^c^
Flavour	7.70 ± 1.56 ^a^	6.50 ± 0.82 ^b^	4.20 ± 1.54 ^c^	6.10 ± 0.87 ^b^	6.00 ± 1.24 ^b^	6.10 ± 1.66 ^b^	6.20 ± 1.03 ^b^
Syneresis	7.90 ± 1.20 ^a^	7.75 ± 1.33 ^a^	4.40 ± 1.42 ^c^	5.40 ± 1.03 ^bc^	5.40 ± 1.07 ^bc^	5.70 ± 1.82 ^b^	5.80 ± 1.35 ^b^
Overall acceptability	7.90 ± 1.79 ^a^	6.80 ± 0.57 ^b^	5.30 ± 0.95 ^d^	5.70 ± 0.69 ^cd^	6.20 ± 0.82 ^bc^	6.40 ± 1.57 ^c^	6.50 ± 1.24 ^bc^

Values are means ± standard deviation of three determinations. Values with different letters (superscript) on the same row are significantly different (*p* ≤ 0.05). CTR—Commercial yoghurt; LCTR—gelatin-stabilized yoghurt; YoCN—yoghurt without stabilizer; YoPN—yoghurt stabilized with native starch; YoPA1—yoghurt stabilized with DS 0.05 acetylated starch; YoPA2—yoghurt with stabilized with DS 0.09 acetylated starch; YoPA3—yoghurt stabilized with DS 0.14 acetylated starch.

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
