# Peer review of "Technological Properties of Acetylated Pigeon Pea Starch and Its Stabilized Set-Type Yoghurt"

_foods, 2020, doi:10.3390/foods9070957_

Round 1

Reviewer 1 Report

The manuscript shows a potential application of using acetylated pigeon pea starch in yogurt. Some concerns rised as follows:

What is the convenience to prepare and use the acetylated pigeon pea starch in comparison to other available starches or food thickening agents? Is there any safety compliance about acetylated pigeon pea starch?

I think the use of functional in the title is not appropriate. I would rather say technological or a something similar, indeed no functional determinations were reported.

- line 32-33: replace “to cause a positive effect in” with “to improve”

- lines 40-43: the sentences should be rewritten in a better way

- line 173: fashion?

- lines 216-218: the sentence is wrong: the yoghurts containing acetylated pea starch had lower syneresis…than what? you should also remark that commercial yogurt had far lower syneresis than any other sample even with a lower solid content.

- lines 232-236: English structure of the sentence has to be corrected

- lines 246-247: it is not clear, which yogurt are talking about? be more specific when you compare something with something else

Discussion: in my opinion the point by point argumentation of all the parameters makes reading less flowing, i would rather be more concise and try to link related reasoning.

page 11, line 8: rearrange the sentence

page 12, line 102: Acidity is synonymous to pH…this is not correct, be more scientific

page 13: lines 110-111: starter culture in yogurt exerts an essential role not only for the acidification but also aromatic compounds, texture and sweetness. When you compare a commercial yogurt to experimental samples, you should consider that many starter cultures are selected for the ability to produce exopolysaccharides with texturizing effects, so the comparison may be valid as indication.

Reviewer 2 Report

The Authors present a study aimed to reduce the syneresis in yogurt samples by the addition of acetylated pigeon pea starch as a stabilizer.

The manuscript is well presented and the results properly discussed and compared with literature.

Some minor issues:

- Please, choose between “yogurt” (as in the title), and yoghurt (as in the rest of the manuscript).

-  Tables: what are meaning the indexes “a, b, c, d, e, f,….”? Please add the explanation at the bottom of the table.

- Line 189: “starches.PPNS:” missing space.

- Improve the general quality of the plots

Reviewer 3 Report

This manuscript is on the preparation and detailed analysis of yogurt products with acetylated pigeon pea starch. A variety of sensory characteristics were tested along with a wide array of technical functionalities.

The paper is technically sound. The findings are of relevance for the readers of the Journal. The paper is rich in data and the presentation of data is mostly acceptable. The work presented in this manuscript has been carefully down and from a very complete perspective. In my opinion, it is acceptable subject to some minor comments. Although I think that this paper will be a useful addition to the current literature, the data explanations should be mechanistically improved.

The major shortcomings include:

L 42 food starches  - specifically received from ...

L50-53 source?

L103 why sugar was added to natural yogurt

L106 DANISCO Yo Mix; 50 mg/L of yoghurt mix - What was the power of starter cultures? Unit activity?

L107 45 ºC for 12 h - please explain why such a long time?

L113 how many times each measurement was repeated; how many liters of yogurt were produced; whether yogurt was made on a laboratory scale or in a dairy

L108 develop proper acidity – specifically?

L110 p. 13 starter culture, milk type, fermentation time and temperaturÄ™ - this was not studied - there were no different variables - please explain

L113 p. 13 - why camel milk appears in the discussion

Reviewer 4 Report

The authors characterized some properties of modified pea starch and studied the functionality of the modified starches in set-yogurt. The topic has value to the industry and academia, however, the manuscript contains serious flaws.

Title: The title is a bit confusing, did the authors focus on functionality of modified starch (in the application of yogurt) or the functionality of yogurt?

Abstract (line 15-19): the authors used yogurt without starch as a reference control for comparisons on WHC and syneresis. This doesn’t make any sense. Although, yogurt without starch may be used as a side control, the manuscript focuses on modified starch, therefore yogurt containing native starch should be used as the primary control sample.

Introduction: the manuscript focused on acetylated pigeon pea starch, however, the authors did not justify the motivation/driving force of doing such study e.g. why native pigeon pea starch cannot stabilized yogurt gel and what works have been done previously to study native pea starch as a stabilizer/texturizer in acidified gel systems. The authors need to introduce more specific technical problems to justify the needs of modification of pea protein especially for the application of yogurt. Also, the authors need to introduce key characteristics of such modified starch.

107: why the fermentation took so long (12h) to achieve proper acidity for the system containing low protein content? This doesn’t relevant to the industry process or other bench top scale R&D works. Was there a contamination in the yogurt milk? The authors need to report how long it took to reach pH 4.6.

105: how yogurt milk was mixed, did the authors applied high shear (homogenization) to the yogurt milk containing modified or native starch granules? The authors stated in the introduction that native starch needs to be modified to improve its shear resistance. The authors did not mention or specify any shear conditions before or after fermentation, therefore, the authors did not test the shear resistant functionality of the modified starch. Moreover, in an industrial manufacturing process of yogurt milk, starch is usually added in fresh milk or reconstituted milk first, then the overall milk system will be gone through a high shear homogenization treatment + a heat treatment. The sample preparation in this work doesn’t mimic or reflect the commercial yogurt processing method in terms of key procedures and processing parameters (e.g. the homogenization part).

111-113: why commercial yogurt was used as a control sample? Commercial yogurt does not relevant to this work at all, and the commercial sample is not even comparable to the samples made by the authors. Because the preparation method of testing samples do not mimic the commercial manufacture process. E.g. commercial (large scale) process uses HTST for heat treatment. Also, the pH was so different in the commercial sample in comparison to the testing samples. Moreover, the authors did not report the composition information for the testing samples.

Section 2.2: the authors did not tell how many replicates were prepared for each sample, it is not clear how many containers of yogurt were prepared for each sample

Section 2.3.4 and 2.3.5: the authors need to provide more practical details regarding the sub-sampling procedure. E.g. was the sub-sample taken from the same bulk sample on each testing day? Or was the sub-sample taken from different new containers.

Table 2: the gelatinized native starch (PPNS) had higher viscosity than modified starch (PPAS3), then why in the sensory analysis YoPN had lower firmness than YoPA3?

Table 4: This paper is about the functionality of starch in yogurt application. Besides sensory analysis, the author should perform instrumental analysis for characterizing the textural properties of yogurt samples (e.g. using texture characterization instrument and/or rheometer). The instrumental techniques may provide relatively more objective quantitative information about the differences in textural attributes. Moreover, I found the data/results in table 4 is odd. E.g. in the “syneresis row”, YoPN (5.40±1.03e) is significantly different than YoPA3 (5.80±1.35c); however, in the “overall acceptability row”, YoPN (5.70±0.69bc) is NOT significantly different than YoPA3 (6.50±1.24b), where the difference of mean values is larger and the difference of standard deviation is smaller than the paired data in the “syneresis row”.

Round 2

Reviewer 1 Report

Authors sufficiently replid to comments and critics arose by reviewers, although did not improve the manuscript elsewhere for a general better reading.

Author Response

Authors sufficiently replied to comments and critics arose by reviewers, although did not improve the manuscript elsewhere for a general better reading.

Response: Thank you.

During our last revision, we edited some sections within the body of the MS. We could not completely re-write the discussion as previously suggested by the reviewer; this is because our style of discussion is due to the fact that majority of the parameters detailed have synergistic effects on the technological properties of the starch.
